# Evolutionary Game Analysis of Power Generation Groups Considering Energy Price Fluctuation

**Yu Jiang** [1,2]**, Changyu Qian** [3] **, Jie Yu** [3,]*****, Luyao Zhou** [4]**, Zheng Wang** [3]**, Qian Chen** [4]**, Yang Wang** [1,2] **and Xiaole Ma** [1]

1 State Grid Jiangsu Electric Power Company Ltd., Nanjing 210000, China
2 Jiangsu Power Exchange Center Co., Ltd., Nanjing 210008, China
3 School of Electrical Engineering, Southeast University, Nanjing 210096, China
4 College of Energy and Electrical Engineering, Hohai University, Nanjing 211100, China
* Correspondence: yujie@seu.edu.cn; Tel.: +86-13951005981

**Abstract:** As the double carbon target continues to be promoted and the installed capacity of gas-fired power generation gradually expands, whether and when gas-fired power generation should enter the market is a major concern for the industry. This paper analyzes the change in power generation cost and the characteristics of bidding behavior of the power generation group with the fluctuation of primary energy price to study the timing and role of gas power generation entering the market. The evolutionary game model for gas- and coal-fired power generation groups based on the influence of multiple factors takes into account factors such as generation costs, the number of power generation groups, generation capacity, supply, and demand. The equilibrium states of the power generation groups under various scenarios and the conditions that need to be satisfied for the equilibrium states are analyzed, and a method for determining the stable equilibrium point of the evolving market game is proposed. Examples use the actual price fluctuations of coal and natural gas as input data to validate the rationality of the paper's model and the stable equilibrium approach.

**Keywords:** fluctuations in energy prices; group evolution game; generator bidding; gas power generation into the market

## 1. Introduction

The development of clean and low-carbon energy is accelerating under China's goal of carbon peaking and carbon neutrality. Natural gas power generation has a high unit heat content, and carbon emissions are nearly half that of coal-fired power generation [1]. In the future, it will be a promising resource for power generation. As seen in developed countries, gas-fired power generation has increasingly become the main source of electricity, avoiding the negative problems associated with other forms of power generation. Therefore, realizing the development of gas-fired power generation is an inevitable choice for the transformation of China's power generation mode. It is expected that by 2020, the country's installed gas-fired power generation capacity will reach around 110 million kilowatts, accounting for 5.5% of the country's total installed power generation capacity [2]. Gas-fired power generation is currently mainly involved in peak shaving. Gas-fired power generation participates in market-based transactions by selecting periods that match its on-grid electricity price and intermittently participates in market-based transactions [3]. With the increase in installed capacity of gas-fired power generation, whether gas-fired power generation should enter the market and compete with coal-fired power generation; how to analyze the timing and role of gas-fired power generation in the market is a hot issue in the power industry at present.

With the continuous promotion of power system reform, all industries and commerce have entered the power market, and the industrial and commercial catalog electricity price has been canceled [4]. The fluctuation of primary energy prices and the uncertainty of supply and demand will greatly affect the bidding behavior and market price of power

generation groups. The price of thermal coal and natural gas is often fluctuated by factors such as temperature deviation, supply shock, and government capacity control [5,6]. How quantifying the impact of primary energy price fluctuations on the behavior of power generators to reasonably evaluate the game operation situation of the power market is a problem that market management should pay attention to.

### 1.1. Research Background

Coal-fired power plants use thermal coal as the primary energy for power generation, and coal prices have a huge impact on their power generation costs and objectively become a coal-fired power generation group. Similarly, gas-fired power plants use natural gas as primary energy and objectively form a gas-fired power generation group. Among the market analysis methods based on game theory, the commonly used mathematical methods included the supply function equilibrium model [7], the Stackelberg model [8], and the Conjectural Variation model [9]. The above models cannot consider the long-term impact of the energy price-generation cost transmission mechanism on the group bidding strategy of power generators, so they are not suitable for studying the group behavior of many power generators in complex market environments such as primary energy price fluctuations.

Evolutionary game is suitable for studying group game behavior in the market. Its theory originates from the long-term interaction of competitive behaviors in the process of biological evolution [10]. The biggest feature of the evolutionary game is that the strategy that produces higher expected returns is used as the direction of the optimal strategy [11,12]. In addition, it has a good prospect in the group behavior characteristics of a long period and a large spatial span. The theory of group evolution strategy usually emphasizes the gradual optimization of the subject's strategy under the influence of direct macro factors, and the influence of various factors on the evolution in a complex environment can be reflected in the payment matrix of the power generation group [13]. Therefore, the evolution process of the bidding behavior on the power generation side can be comprehensively and accurately analyzed when the energy price fluctuates.

Under the background of limited rationality and information asymmetry of market participants, evolutionary game theory has been widely used in the study of market players' behavioral decision-making problems in recent years. Based on the complexity of the real market environment, many studies tend to use multiple game models to simulate the behavior of market participants. For example, Tan J et al. proposed a hierarchical game approach for enabling reliability-differentiated services in a residential distribution network with plug-in hybrid electric vehicles. At the upper level of the hierarchical game, an evolutionary game is formulated to optimize the management of the vehicle-to-grid capacity of each vehicle. Ref. [14] Paudel A at al. Proposed a novel game-theoretic model for peer-to-peer (P2P) energy trading among the prosumers in a community, in which evolutionary game theory is used to model the dynamics of the buyers for selecting sellers. Ref. [15] Chai B et al. studied the demand response problem of multiple utilities and multiple residential users based on evolutionary games and non-cooperative games, and the results show that the proposed scheme can significantly reduce peak loads. Ref. [16] In addition, the evolutionary game has been involved in power generation bidding [17], energy system supervision [18,19], microgrid energy management [20], and other fields.

The above references do not consider the long-term dynamic interaction behavior of various types of power generators under the background of primary energy price fluctuations. By considering the impact of primary energy price fluctuations on power generation costs, this paper proposes an evolutionary trend analysis of bidding for coal-fired and gas-fired power generation groups based on evolutionary games and establishes a power generation side game model under the influence of multiple factors. Under the background of primary energy fluctuation, the equilibrium state of the power generation side game under different supply-demand relations and the conditions that the equilibrium state needs to meet are proposed. The cases use the actual price fluctuations of coal and

natural gas as input data to verify the rationality of the model and the stable equilibrium method in this paper.

*1.2. Main Work and Innovations*

Where coal-fired generation capacity can meet load requirements and natural gas prices are higher than coal, it is not appropriate for the gas-fired generation to enter the market and bid for the same, making it act as a peaking unit or participate in the ancillary services market is the solution to maintain stable system operation and facilitate market balancing. As new coal-fired units are banned, and more and more coal-fired units are due to be taken out of service, it is reasonable for gas-fired units to enter the market and take on the task of supplying electricity when coal-fired generation capacity cannot fully support the electricity load.

The main work of this paper is to deduce the conditions under which the primary energy price ratio between natural gas and power coal can be set to meet the market equilibrium and not push up the market price of electricity when coal-fired generation cannot meet the load demand; otherwise, the market equilibrium can be achieved, and the stability of electricity price can be maintained by setting separate bids for coal-fired generation groups.

In terms of the organization and structure of this paper, an evolutionary game model for the group of power producers under the influence of multiple factors is firstly developed. Secondly, the cost of electricity generated by the group of power producers under the fluctuation of primary energy prices, including the cost of electricity generated by coal and the cost of electricity generated by gas, is analyzed. Then, the multifactor game analysis and the dynamic evolutionary game analysis of the group of power producers under the influence of energy price fluctuations are studied. Finally, a simulation is carried out to analyze the price fluctuations in the energy market and the cost of electricity and to analyze the bidding game of the group of power producers considering the fluctuations in primary energy prices.

This paper discusses the application of the evolutionary game model to the simulation and analysis of generation-side bidding dynamics in the electricity market. The evolutionary path and game equilibrium of the evolutionary game under multiple scenarios and multiple initial conditions are discussed. The focus of this paper is not on proposing new bidding strategies but on simulating the game equilibrium of the market as a whole when market players adopt different bidding strategies, to determine whether the declared electricity on the generation side of the market and the declared electricity on the consumer side are in balance. Based on the classical evolutionary game model in Ref. [11], the contribution of this paper is to model the bidding problem for different types of generating units in the electricity market and then perform a swarm simulation using the evolutionary model theory, which extends the application of evolutionary game models in complex environments and analyses the game paths and game equilibria in multiple scenarios. The paper designs and validates the electricity market bidding mechanism based on the evolutionary game path and evolutionary game results.

## 2. Construction of Evolutionary Game Model of Power Generator Group under the Influence of Multiple Factors

In the electricity market, the power generation group participating in the bidding game sets the bidding strategy according to the power generation group's primary energy price, power generation efficiency, market supply and demand, and other factors. The subjects of the bidding game all show bounded rationality, which will be affected by incomplete transaction information and the limited thinking of market transaction participants. With the fluctuation of natural gas and thermal coal prices in the primary energy market, coal-fired and gas-fired power generation groups will adjust their bidding strategies according to their respective power generation costs to maximize revenue as much as possible.

Power producers purchase gas or coal from the primary energy market for storage for power generation. Power producers, large industrial consumers, and electricity sellers participate in bidding in the electricity market and then conduct electricity exchange. The specific clearing electricity quantity and electricity price are determined by the market mechanism.

When the price of the primary energy market fluctuates, the clearing price of the electricity market will fluctuate due to the change in the power generation cost of the power suppliers. According to evolutionary game theory, the power market management department can simulate the bidding trend of coal-fired power generation units and gas-fired power generation units, respectively, and analyze their optimal bidding strategies. Then adjust the bidding electricity of the two types of generating units to change the electricity structure on the generation side and avoid sharp fluctuations in the transaction price. The bidding strategies of the power generation group in the electricity market are divided into two types, one type is cost-based bidding based on the price of primary energy, and the other type is strategic bidding that pursues high-profit margins. Cost-based bidding means that the bidding price is lower, and more electricity will be cleared; Strategic bidding means a higher bidding price and more profits. Suppose the two bidding strategies of coal-fired power generation groups are cost bidding $P_{coal}^L$ and strategic bidding $P_{coal}^H$, and the proportion of power generation groups who choose these two strategies are $p$ $1-p$ ($0 \leq p \leq 1$); The two bidding strategies of the gas-fired power generation group are cost bidding $P_{gas}^L$ and strategic bidding $P_{gas}^H$, and the proportions of power generators who choose these two strategies are $q, 1-q$ ($0 \leq q \leq 1$). The payoff matrix of the evolutionary game of market competition among power generation groups is shown in Table 1. Table 1 $u_i$ and $v_i$ ($i = 1, 2, 3, 4$) denote the revenue from coal-fired power generators and gas-fired power generators, respectively.

**Table 1.** Power market operation situation system dynamics model module division.

| Group Bidding Strategy of Coal-Fired Power Generators | Group Bidding Strategy of Gas-Fired Power Generators | |
|---|---|---|
| | $P_{gas}^L(q)$ | $P_{gas}^H(1-q)$ |
| $P_{coal}^L(p)$ | $(u_1, v_1)$ | $(u_2, v_2)$ |
| $P_{coal}^H(1-p)$ | $(u_3, v_3)$ | $(u_4, v_4)$ |

Based on the above game model assumptions, the gas-fired generation group's expected revenue when bidding based on cost is, as described in Equation (1).

$$E_{gas}^L = pv_1 + (1-p)v_3 \tag{1}$$

Based on the strategic tender, the expected revenue of the gas-fired generation group is shown in Equation (2)

$$E_{gas}^H = pv_2 + (1-p)v_4 \tag{2}$$

The average expected revenue of the gas-fired generation group is shown in Equation (3)

$$\overline{E}_{gas} = qE_{gas}^L + (1-q)E_{gas}^H \tag{3}$$

Based on the cost bidding, the expected revenue of the coal-fired generation group is as described in Equation (4)

$$E_{coal}^L = qu_1 + (1-q)u_2 \tag{4}$$

The expected revenue of the coal-fired generation group based on the strategic tender is shown in Equation (5)

$$E_{coal}^H = qu_3 + (1-q)u_4 \tag{5}$$

The average expected revenue of the coal-fired generation group is shown in Equation (6)

$$\overline{E}_{coal} = pE_{coal}^L + (1-p)E_{coal}^H \tag{6}$$

Based on the set of equations in Equations (1)–(6), the replicated dynamic differential equations for coal-fired and gas-fired generating units are shown in Equations (7) and (8) as follows.

$$\dot{p} = \frac{dp}{dt} = p(1-p)[(u_2 - u_4) + q(u_1 - u_3 - u_2 + u_4)] \tag{7}$$

$$\dot{q} = \frac{dq}{dt} = q(1-q)[(v_3 - v_4) + p(v_1 - v_3 - v_2 + v_4)] \tag{8}$$

Therefore, the evolutionary bidding state on the power generation side can be described by Equations (7) and (8). Let Equations (7) and (8) be equal to 0. It can be solved to its local equilibrium point in space $N\{(p,q); 0 \le p,q \le 1\}$. Then, there are five equilibrium points in the system composed of evolutionary game models, which is $E_1(1,1)$, $E_2(1,0)$, $E_3(0,1)$, $E_4(1,1)$, $E_5(p_0, q_0)$. Coordinates $p_0$ and $q_0$ are calculated according to Equations (9) and (10):

$$p_0 = (v_4 - v_3)/(v_1 - v_2 - v_3 + v_4) \tag{9}$$

$$q_0 = (u_4 - u_2)/(u_1 - u_3 - u_2 + u_4) \tag{10}$$

For a population dynamic described by a system of differential equations, the stability of its equilibrium point is obtained by the local stability analysis of the Jacobi matrix. The Jacobi matrix of the above evolutionary game system is described in Equation (11):

$$J = \begin{pmatrix} \frac{\partial \dot{p}}{\partial p} & \frac{\partial \dot{p}}{\partial q} \\ \frac{\partial \dot{q}}{\partial p} & \frac{\partial \dot{q}}{\partial q} \end{pmatrix} \tag{11}$$

where in, the calculation formula of each element in the Jacobian matrix is shown in Equation (12).

$$\begin{cases} \frac{\partial \dot{p}}{\partial p} = (1-2p)[(u_1 - u_2 - u_3 + u_4)q + u_2 - u_4] \\ \frac{\partial \dot{p}}{\partial q} = p(1-p)(u_1 - u_2 - u_3 + u_4) \\ \frac{\partial \dot{q}}{\partial p} = q(1-q)(v_1 - v_2 - v_3 + v_4) \\ \frac{\partial \dot{q}}{\partial q} = (1-2q)[(v_1 - v_2 - v_3 + v_4)p + v_3 - v_4] \end{cases} \tag{12}$$

For an evolutionary game with two agents and two strategies, judging whether an equilibrium point is stable requires calculating the determinant and trace of its Jacobi matrix. If the determinant of an equilibrium point is greater than zero and the trace is less than zero, the equilibrium point is asymptotically stable and is a stable equilibrium point (SEP); If both the determinant and the trace are greater than zero, the equilibrium point is unstable and is an unstable equilibrium point (UEP); If the determinant is less than zero, the local equilibrium point is a saddle point (SP), that is, the system is in a critical evolutionary state at this point, and is still in an unstable equilibrium state.

The next section will analyze the stability of each strategic equilibrium point of coal-fired and gas-fired power generation groups based on the fluctuation of primary energy prices and the influence of different market supply and demand.

## 3. Analysis of Power Generation Cost under Primary Energy Price Fluctuation

In the context of market-oriented competition, coal-fired and gas-fired power generation groups must first consider their power generation costs when participating in online bidding. The fluctuation of primary energy price has a significant impact on the marginal cost of the power generation group, and the change in the power generation cost will directly affect the bidding decision of the power generation group. The following section

will analyze the impact of energy price fluctuations on power generation costs from the perspective of the cost structure of the power generation group.

### 3.1. Coal-Fired Power Generation Cost Analysis

The power generation cost of coal-fired power generation should first consider the spot price of thermal coal, transportation costs, and tax rates, and calculate the arrival price of thermal coal; then calculate the variable cost of coal-fired power generation; finally, calculate the marginal cost of coal-fired power generation. It includes the following steps:

First, the power coal arrival price is derived from the power coal spot price, as described in Equation (13).

$$\lambda_{faccoal} = [\lambda_{coal}\eta_{coal}(1 - T_{coal}) + TR_{coal}(1 - T_{trcoal})] \tag{13}$$

In the formula: $\lambda_{faccoal}$ is the arrival price of thermal coal; $\lambda_{coal}$ is the spot price of thermal coal; $\eta_{coal}$ is the standard coal conversion rate; $T_{coal}$ is the thermal coal tax rate; $TR_{coal}$ is the coal transportation cost; $T_{trcoal}$ is the coal transportation tax rate.

The variable cost of coal-fired electricity generation is then calculated based on the arrival price of power coal, as shown in Equation (14).

$$C_{coal}^{VAR} = (\lambda_{faccoal}\sigma_{coal} + C_{wat} + C_{env})(1 + T_{coalelec}) \tag{14}$$

In the formula: $C_{coal}^{VAR}$ is the variable cost of coal-fired power generation; $\sigma_{coal}$ is the comprehensive coal consumption; $C_{wat}$ is the water production cost; $C_{env}$ is the environmental cost; $T_{coalelec}$ is the coal-fired power generation tax rate.

Finally, the marginal cost of coal-fired power generation can be obtained from the variable cost of coal-fired power generation, as shown in the following Equation (15):

$$C_{coal} = (C_{coal}^{VAR} + C_{coal}^{Const})(1 + \eta_{cofacelec}) \tag{15}$$

In the formula: $C_{coal}$ is the marginal cost of coal-fired power generation; $C_{coal}^{Const}$ is the fixed cost of coal-fired power generation; $\eta_{cofacelec}$ is the power consumption rate of coal-fired power plants.

### 3.2. Gas-Fired Power Generation Cost Analysis

The power generation cost of gas-fired power generation needs to first consider its transportation costs and tax rates from the CIF/ex-factory price of natural gas to calculate the arrival price of natural gas; then calculate the variable cost of gas-fired power generation; finally, calculate its marginal cost of gas-fired power generation. It includes the following steps:

First, the arrival price of natural gas is obtained from the CIF/ex-factory price of natural gas, as shown in the following Equation (16):

$$\lambda_{facgas} = \lambda_{gas}(1 - T_{gas}) + TR_{gas}(1 - T_{trgas}) \tag{16}$$

In the formula: $\lambda_{facgas}$ is the arrival price of natural gas; $\lambda_{gas}$ is the CIF/ex-factory price of natural gas; $T_{gas}$ is the natural gas tax rate; $TR_{gas}$ is the natural gas transportation cost; $T_{trgas}$ is the natural gas transportation tax rate.

Then the variable cost of gas-fired power generation can be obtained from the arrival price of natural gas, as shown in the following Equation (17):

$$C_{gas}^{VAR} = \lambda_{facgas}\sigma_{gas}(1 + T_{gaselec})/(1 - \eta_{gafacelec}) \tag{17}$$

In the formula: $C_{gas}^{VAR}$ is the variable cost of gas-fired power generation; $\sigma_{gas}$ is the gas power consumption; $\eta_{gafacelec}$ is the power consumption rate of gas-fired power plants; $T_{gaselec}$ is the gas-fired power generation tax rate.

Finally, the marginal cost of gas-fired power generation can be obtained from the variable cost of gas-fired power generation, as shown in the following Equation (18):

$$C_{gas} = C_{gas}^{VAR} + C_{gas}^{Const}(1 + \eta_{gafacelec})$$ (18)

In the formula: $C_{gas}$ is the marginal cost of gas-fired power generation; $C_{gas}^{Const}$ is the fixed cost of gas-fired power generation.

Therefore, the influence of energy price on power generation cost can be calculated by Formula (13)–(18). The following will further analyze the bidding game behavior and evolution trend of gas and coal-fired power generation groups under the influence of supply-demand relationship changes and cost fluctuations.

## 4. Analysis of the Bidding Game of Power Generation Groups under the Fluctuation of Primary Energy Price

The fluctuation of primary energy prices, thermal coal prices, and natural gas prices determine the power generation cost of the two groups of power generators, thus affecting their bidding strategies and causing the entire market to evolve to different game equilibrium points. In addition, the supply and demand situation will also have a significant impact on the bidding strategy of the power generation group.

First, it is assumed that the single-unit power generation capacity, power generation cost, and the number of power generation groups of coal-fired power generation groups are $Q_{coal,i}$, $C_{coal}$ and $m$ respectively, the gas-fired power generation group is assumed to be $Q_{gas,i}$, $C_{gas}$ and $n$ respectively. Then the power generation capacity participating in the market competition is $Q_{SP} = \sum_{i=1}^{m} Q_{coal,i} + \sum_{i=1}^{n} Q_{gas,i}$, and the total demand for the electricity market is $Q_{DM}$.

In addition, let the cost bidding coefficient of coal-fired power generation groups be $\mu_{coal}$ the strategic bidding coefficient $v_{coal}$. Similarly, let the cost bidding coefficient of gas-fired power generation groups be $\mu_{gas}$ the strategic bidding coefficient is $v_{gas}$, and assume $\mu_{coal} = \mu_{gas}$ $v_{coal} = v_{gas}$.

Based on this, according to the total market demand for power generation $Q_{DM}$ and the power generation capacity participating in market competition $Q_{SP}$, it can be divided into the following two typical situations under the influence of different supply and demand relationships and the fluctuation of primary energy prices:

Scenario 1: $\sum_{i=1}^{m} Q_{coal,i} < Q_{DM}$, $\sum_{i=1}^{n} Q_{gas,i} < Q_{DM}$, $\sum_{i=1}^{m} Q_{coal,i} + \sum_{i=1}^{n} Q_{gas,i} \geq Q_{DM}$, the bidding power on the generation side exceeds bidding power on the consumption side, and the power generation of coal-fired and gas-fired power generators participating in the market is each smaller than the market demand, and the total power generation of coal-fired and gas-fired power generators participating in the market is greater than the market demand;

Scenario 2: $\sum_{i=1}^{m} Q_{coal,i} \geq Q_{DM}$, $\sum_{i=1}^{n} Q_{gas,i} \leq Q_{DM}$, the market supply exceeds the demand, and the coal-fired power generation group participating in the market generates more power than the market demand, and the gas-fired power generation group participating in the market generates less power than the market demand.

In order to represent the change in the cost of electricity generation caused by fluctuations in energy prices, the cost ratio of coal and gas is introduced as $\varepsilon$, let $\varepsilon = C_{coal}/C_{gas}$, for the various situations mentioned above, this section will discuss the equilibrium points' stable situation of the two groups of power generators when $0 < \varepsilon < 1$ (i.e., the cost of coal-fired power generation is lower than that of gas) and $\varepsilon > 1$ (i.e., the cost of coal-fired power generation is higher than that of gas).

The four cases of $0 < \varepsilon < 1$ and $\varepsilon > 1$ in the two scenarios are discussed below.

Scenario 1: $\sum_{i=1}^{m} Q_{coal,i} < Q_{DM}, \sum_{i=1}^{n} Q_{gas,i} < Q_{DM}, \sum_{i=1}^{m} Q_{coal,i} + \sum_{i=1}^{n} Q_{gas,i} \geq Q_{DM}$, the bidding power on the generation side exceeds bidding power on the consumption side, the power generation of coal-fired and gas-fired power generators participating in the market is smaller than the market demand, and the total power generation of coal-fired and gas-fired power generators participating in the market is greater than the market demand;

Scenario 1.1: $C_{coal} = \varepsilon C_{gas}, 0 < \varepsilon < 1$ that is, the cost of coal-fired power generation is lower than that of gas-fired. The payment parameters for this situation are derived as follows:

When both coal-fired and gas-fired power generation groups choose cost bidding, the calculation formulas $u_1 \ v_1$ are shown in Equation (19).

$$\begin{cases} u_1 = P_{coal}^L - C_{coal} = (\mu_{coal} - 1)C_{coal} \\ v_1 = \dfrac{Q_{DM} - \sum_{i=1}^{m} Q_{coal,i}}{\sum_{i=1}^{n} Q_{gas,i}}(P_{gas}^L - C_{gas}) = \dfrac{Q_{DM} - \sum_{i=1}^{m} Q_{coal,i}}{\sum_{i=1}^{n} Q_{gas,i}}(\mu_{gas} - 1)C_{gas} \end{cases} \tag{19}$$

When the coal-fired power generation group chooses cost bidding and the gas-fired power generation group chooses strategic bidding, the calculation formulas $u_2 \ v_2$ are shown in Equation (20):

$$\begin{cases} u_2 = P_{coal}^L - C_{coal} = (\mu_{coal} - 1)C_{coal} \\ v_2 = \dfrac{Q_{DM} - \sum_{i=1}^{m} Q_{coal,i}}{\sum_{i=1}^{n} Q_{gas,i}}(P_{gas}^H - C_{gas}) = \dfrac{Q_{DM} - \sum_{i=1}^{m} Q_{coal,i}}{\sum_{i=1}^{n} Q_{gas,i}}(v_{gas} - 1)C_{gas} \end{cases} \tag{20}$$

When the gas-fired power generation group chooses cost bidding and the coal-fired power generation group chooses strategic bidding, $P_{coal}^H = v_{coal}C_{coal} = v_{coal}\varepsilon C_{gas}$; $P_{gas}^L = \mu_{gas}C_{gas}$. At this time, the quotation of coal-fired and gas-fired power generation groups depends on the relationship between $\varepsilon$ and $\mu_{gas}/v_{coal}$ the two situations will be discussed as follows.

If $\varepsilon < \mu_{gas}/v_{coal}$, then $P_{coal}^H < P_{gas}^L$, at this time, the price quoted by the strategic bidding of the coal-fired power generation group is still lower than the price quoted by the cost bidding of the gas-fired power generation group, the payment parameters of the two types of generation groups can be expressed as Equation (21).

$$\begin{cases} u_3 = P_{coal}^H - C_{coal} = (v_{coal} - 1)C_{coal} \\ v_3 = \dfrac{Q_{DM} - \sum_{i=1}^{m} Q_{coal,i}}{\sum_{i=1}^{n} Q_{gas,i}}(P_{gas}^L - C_{gas}) = \dfrac{Q_{DM} - \sum_{i=1}^{m} Q_{coal,i}}{\sum_{i=1}^{n} Q_{gas,i}}(\mu_{gas} - 1)C_{gas} \end{cases} \tag{21}$$

If $\varepsilon > \mu_{gas}/v_{coal}$, then $P_{coal}^H > P_{gas}^L$, at this time, the price quoted by the strategic bidding of the coal-fired power generation group is higher than the price quoted by the cost bidding of the gas-fired power generation group, the payment parameters of the two types of generation groups can be expressed as Equation (22):

$$\begin{cases} u_3 = \dfrac{Q_{DM} - \sum_{i=1}^{n} Q_{gas,i}}{\sum_{i=1}^{m} Q_{coal,i}}(P_{coal}^H - C_{coal}) = \dfrac{Q_{DM} - \sum_{i=1}^{n} Q_{gas,i}}{\sum_{i=1}^{m} Q_{coal,i}}(v_{coal} - 1)C_{coal} \\ v_3 = P_{gas}^L - C_{gas} = (\mu_{gas} - 1)C_{gas} \end{cases} \tag{22}$$

When both coal-fired and gas-fired power generation groups choose strategic bidding $u_4$ and $v_4$ are calculated by Equation (23):

$$\begin{cases} u_4 = P_{coal}^H - C_{coal} = (\nu_{coal} - 1)C_{coal} \\ v_4 = \dfrac{Q_{DM} - \sum\limits_{i=1}^{m} Q_{coal,i}}{\sum\limits_{i=1}^{n} Q_{gas,i}}(P_{gas}^H - C_{gas}) = \dfrac{Q_{DM} - \sum\limits_{i=1}^{m} Q_{coal,i}}{\sum\limits_{i=1}^{n} Q_{gas,i}}(\nu_{gas} - 1)C_{gas} \end{cases} \tag{23}$$

Based on this, when the gas-fired power generation group chooses cost bidding and the coal-fired power generation group adopts strategic bidding, there will be two situations in the payment matrix. The first situation will be discussed below:

Scenario 1.1-a: the price quoted by the strategic bidding of the coal-fired power generation group is lower than the price quoted by the cost bidding of the gas-fired power generation group ($P_{coal}^H < P_{gas}^L$)$\varepsilon < \mu_{gas}/\nu_{coal}$. At this time, the replicator dynamics equation is Equation (24):

$$\begin{cases} \dot{p} = \dfrac{dp}{dt} = p(1-p)(\mu_{coal} - \nu_{coal})C_{coal} \\ \dot{q} = \dfrac{dq}{dt} = q(1-q)\dfrac{Q_{DM} - \sum\limits_{i=1}^{m} Q_{coal,i}}{\sum\limits_{i=1}^{n} Q_{gas,i}}(\mu_{gas} - \nu_{gas})C_{gas} \end{cases} \tag{24}$$

Each element in the corresponding Jacobi matrix is calculated by Equation (25):

$$\begin{cases} \dfrac{\partial \dot{p}}{\partial p} = (1-2p)(\mu_{coal} - \nu_{coal})C_{coal} \\ \dfrac{\partial \dot{p}}{\partial q} = 0 \\ \dfrac{\partial \dot{p}}{\partial q} = 0 \\ \dfrac{\partial \dot{q}}{\partial q} = (1-2q)\dfrac{Q_{DM} - \sum\limits_{i=1}^{m} Q_{coal,i}}{\sum\limits_{i=1}^{n} Q_{gas,i}}(\mu_{gas} - \nu_{gas})C_{gas} \end{cases} \tag{25}$$

At this time, the equilibrium point E5 is not located in the space range $[0,1] \times [0,1]$. Thus, there are four equilibrium points in this evolutionary game. According to the Jacobi matrix Formula (25) in this Scenario, the stability of each equilibrium point is shown in Table 2.

**Table 2.** Local equilibrium point stability analysis.

| Local Equilibrium Point | Determinant Sign, Trace Sign, Stability in Condition 1 | | | Determinant Sign, Trace Sign, Stability in Condition 2 | | |
|---|---|---|---|---|---|---|
| E1(0,0) | + | - | SEP | + | - | SEP |
| E2(0,1) | - | - | SP | - | + | SP |
| E3(1,0) | - | + | SP | - | - | SP |
| E4(1,1) | + | + | UEP | + | + | UEP |

Condition 1 is as stated in inequality (26):

$$(\mu_{coal} - \nu_{coal})C_{coal} < \dfrac{Q_{DM} - \sum\limits_{i=1}^{m} Q_{coal,i}}{\sum\limits_{i=1}^{n} Q_{gas,i}}(\mu_{gas} - \nu_{gas})C_{gas} \tag{26}$$

Condition 2 is as stated in inequality (27):

$$(\mu_{coal} - v_{coal})C_{coal} > \frac{Q_{DM} - \sum\limits_{i=1}^{m} Q_{coal,i}}{\sum\limits_{i=1}^{n} Q_{gas,i}} (\mu_{gas} - v_{gas})C_{gas} \tag{27}$$

It can be seen from Table 2 that in this scenario, the two power generation groups gradually approach E1(0,0). That is, all of them adopt strategic bidding.

The dynamic evolution trend under condition 1 is shown in Figure 1a, the dynamic evolution trend under condition 2 is shown in Figure 1b, and the arrows in the figure represent the evolution path.

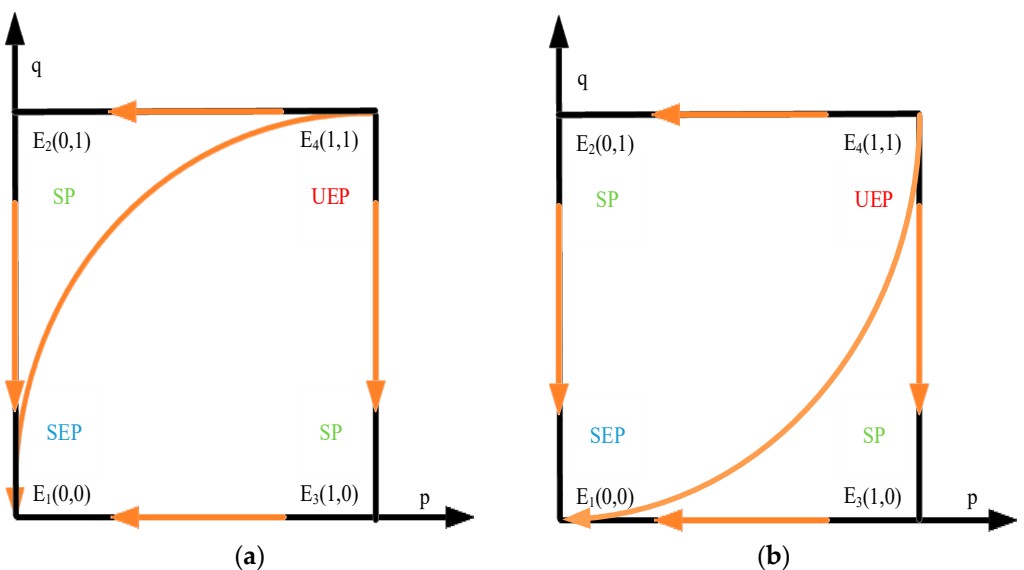

**Figure 1.** The evolution trend of power generation bidding under scenario 1.1-a. (**a**) Trend diagram of the dynamic evolution under condition 1, (**b**) Trend diagram of the dynamic evolution under condition 2.

In scenarios 1 and 2, the derivation of the payment parameters, the replicator dynamics equation and the Jacobi matrix, the stability analysis of each equilibrium point, and the analysis of the evolution trend of the bidding group of power generators in various other conditions are similar to the discussion process of scenario 1.1(a). The detailed derivation process of other scenarios will not be repeated here but only give a summary of the equilibrium point stability analysis and evolution dynamics of all cases, as shown in Tables 3 and 4 and Figure 2.

**Table 3.** Local equilibrium point stability analysis of scenario 1.

| Scenarios | Determinant Sign, Trace Sign, Stability of E1 | | | Determinant Sign, Trace Sign, Stability of E2 | | | Determinant Sign, Trace Sign, Stability of E3 | | | Determinant Sign, Trace Sign, Stability of E4 | | | Determinant Sign, Trace Sign, Stability of E5 | | |
|---|---|---|---|---|---|---|---|---|---|---|---|---|---|---|---|
| 1.1(a)case1 | + | - | SEP | - | - | SP | - | + | SP | + | + | UEP | non-existent | | |
| 1.1(a)case2 | + | - | SEP | - | + | SP | - | - | SP | + | + | UEP | non-existent | | |
| 1.1(b)case1 | + | - | SEP | - | - | SP | - | + | SP | + | + | UEP | non-existent | | |
| 1.1(b)case2 | - | - | SP | + | - | SEP | - | + | SP | + | + | UEP | non-existent | | |
| 1.1(b)case3 | - | - | SP | - | - | SP | - | + | SP | - | + | SP | - | 0 | SP |

**Table 3.** *Cont.*

| Scenarios | Determinant Sign, Trace Sign, Stability of E1 | | | Determinant Sign, Trace Sign, Stability of E2 | | | Determinant Sign, Trace Sign, Stability of E3 | | | Determinant Sign, Trace Sign, Stability of E4 | | | Determinant Sign, Trace Sign, Stability of E5 | | |
|---|---|---|---|---|---|---|---|---|---|---|---|---|---|---|---|
| 1.2(a)case1 | + | - | SEP | - | + | SP | - | - | SP | + | + | UEP | non-existent | | |
| 1.2(a)case2 | - | - | SP | - | + | SP | + | - | SEP | + | + | UEP | non-existent | | |
| 1.2(a)case3 | - | + | SP | - | + | SP | - | - | SP | - | - | SP | - | 0 | SP |
| 1.2(b)case1 | + | - | SEP | - | - | SP | - | + | SP | + | + | UEP | non-existent | | |
| 1.2(b)case2 | + | - | SEP | - | + | SP | - | - | SP | + | + | UEP | non-existent | | |

**Table 4.** Local equilibrium point stability analysis of scenario 2.

| Scenarios | Determinant Sign, Trace Sign, Stability of $E_1$ | | | Determinant Sign, Trace Sign, Stability of $E_2$ | | | Determinant Sign, Trace Sign, Stability of $E_3$ | | | Determinant Sign, Trace Sign, Stability of $E_4$ | | | Determinant Sign, Trace Sign, Stability of $E_5$ | | |
|---|---|---|---|---|---|---|---|---|---|---|---|---|---|---|---|
| 2.1(a)case1 | 0 | - | SP | 0 | - | SP | 0 | + | UEP | 0 | + | UEP | non-existent | | |
| 2.1(b)case1 | - | UK * | SP | + | - | SEP | 0 | + | UEP | 0 | + | UEP | non-existent | | |
| 2.1(b)case2 | - | UK * | SP | - | UK * | SP | 0 | + | UEP | 0 | - | SP | - | 0 | SP |
| 2.2(a)case1 | + | - | SEP | - | UK * | SP | + | + | UEP | - | UK * | SP | non-existent | | |
| 2.2(a)case2 | - | UK * | SP | - | UK * | SP | - | UK * | SP | - | UK * | SP | - | 0 | SP |
| 2.2(b)case1 | + | - | SEP | - | - | SP | - | + | SP | + | + | UEP | non-existent | | |
| 2.2(b)case2 | + | - | SEP | - | + | SP | - | - | SP | + | + | UEP | non-existent | | |

\* The sign "UK" represents "UNKNOWN".

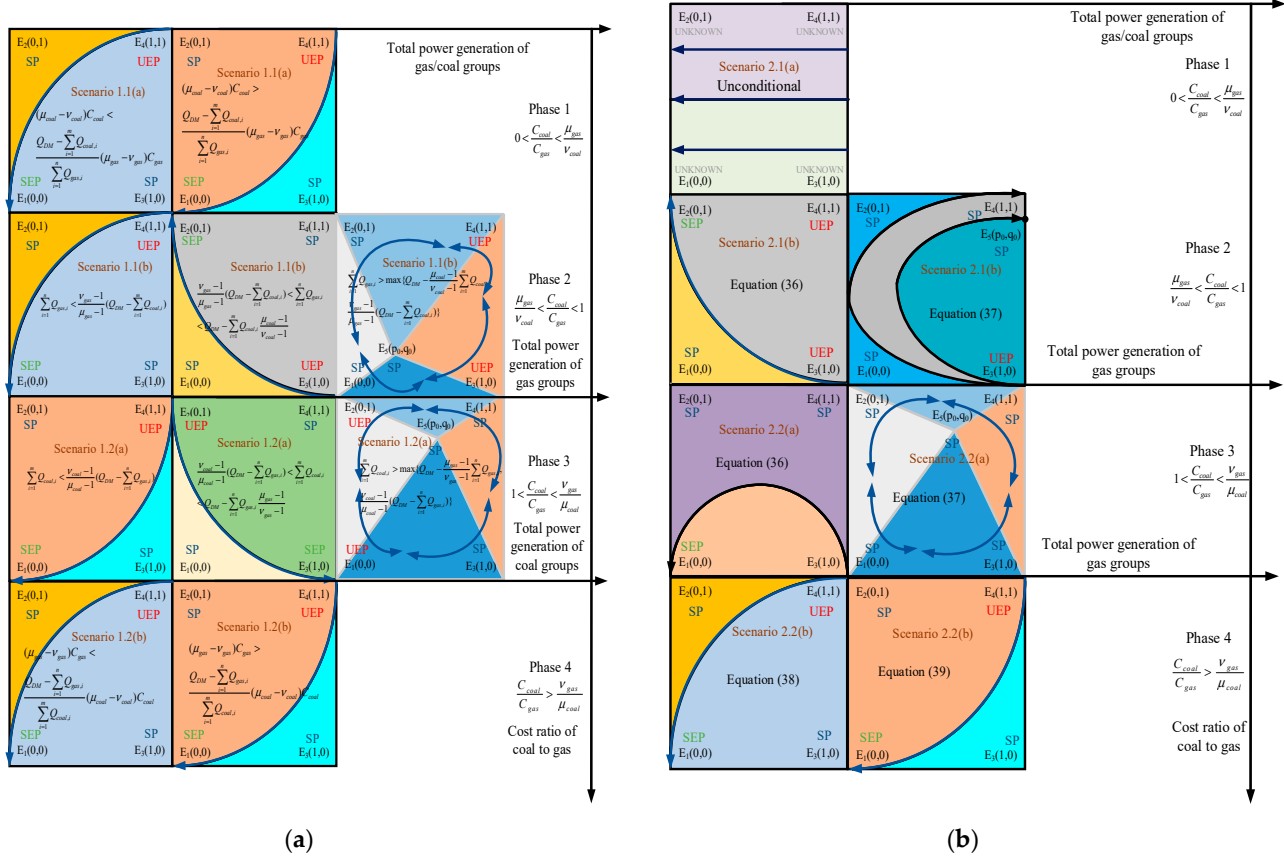

**Figure 2.** The evolution trend of bidding strategies of power generation groups. (**a**) The evolution trend in scenario 1, (**b**) The evolution trend in scenario 2.

## 5. Case Analysis

As China's gas-fired installed capacity gradually increases, and because its carbon emissions are smaller than coal-fired units, it is a trend for gas-fired power to gradually enter the market. However, when gas-fired power generation enters the market and whether it affects the market equilibrium will be a key issue for market management agencies to study and analyze. Firstly, it is necessary to analyze the supply-demand relationship in the electricity market and whether the coal-fired power generation capacity in the market can meet the load demand; Secondly, it is also necessary to analyze the impact of price fluctuations of primary energy sources such as coal and natural gas on power generation costs, to further analyze the game behavior of different power generation groups and deduce market equilibrium conditions. From this, the analysis draws the appropriate time for gas-fired power generation to enter the market, as well as the reasonable market structure setting of whether gas-fired power generation is bidding on the same stage with that as coal-fired.

### 5.1. Primary Energy Price Fluctuation and Power Generation Cost Analysis

In recent years, affected by multiple factors, the price of primary energy in China has always been in a state of high volatility. The trend of coal prices from May 2020 to June 2021 is shown in Figure 3 [21–23]. The coal price index adopts the Yangtze River Steam-Coal Price Index (YRSPI), Ordos Steam-Coal Price Index (OSPI), and Bohai-Rim Steam-Coal Spot Price Index (BSSPI). Affected by the international epidemic, the phased mismatch between supply and demand has led to greater volatility in the thermal coal market in the past year than before. As can be seen from Figure 3, the overall coal price remained relatively stable before the winter of 2020, and the peak season of winter coal consumption in November and December combined with tight market supply, making thermal coal prices continue to rise; in early 2021, with the implementation of national control measures, the bearish market sentiment is strong, causing coal prices to turn around and fall. Since March, the economic situation has continued to improve, and energy demand has increased. The construction of infrastructure projects in various places has been started one after another, and the power consumption of cement and steel coal has been strong, causing the market price of thermal coal to continue to rise.

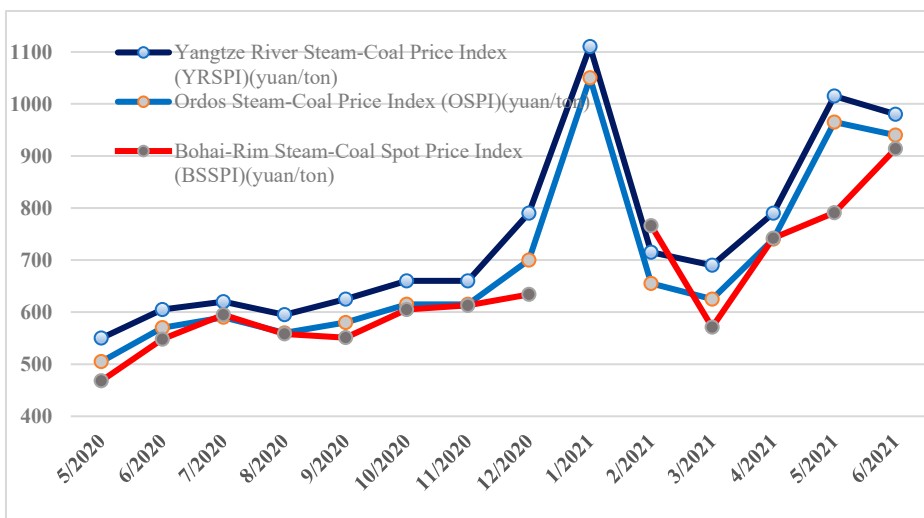

**Figure 3.** May 2020–June 2021 Coal price trend.

The natural gas price trend from May 2020 to June 2021 is shown in Figure 4 [24,25]. The natural gas price has strong seasonal characteristics. In July and August, the domestic heating demand decreased, natural gas entered the off-season of consumption, and the ex-factory price status was stable. Gas prices rise to annual highs during the winter peak of

gas consumption. The recent international epidemic has led to a decline in global natural gas purchasing enthusiasm, and the CIF price of natural gas has fallen steadily.

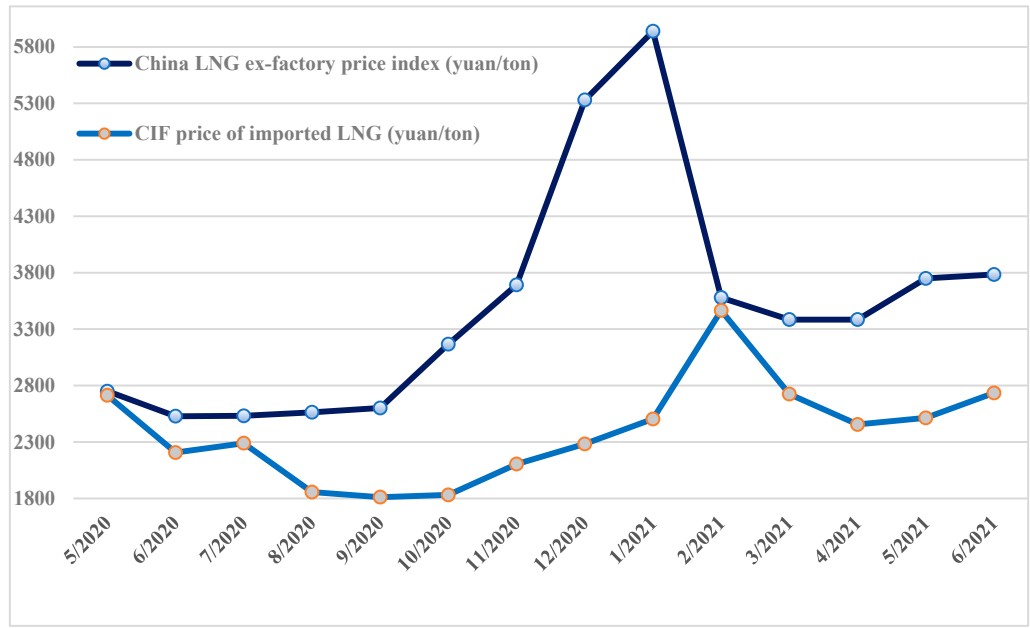

**Figure 4.** May 2020–June 2021 Gas price trend.

This paper takes the coastal areas of the Yangtze River Delta as the research background and selects the Yangtze River Steam-Coal Price Index and CIF price of imported LNG as the primary energy price for coal-fired and gas-fired power generation groups. The tax rate and other energy parameters are set, as shown in Table 5. From Equations (18)–(23) and Table 5, the trend of coal-fired and gas-fired power generation costs from July 2020 to June 2021 can be obtained, as shown in Figure 5.

**Table 5.** Parameter settings of the tax rate, freight, and power plant.

| Coal-Fired Power Generation Group | Parameter Settings | Gas-Fired Power Generation Group | Parameter Settings |
|---|---|---|---|
| Standard coal conversion rate $\eta_{coal}$ (%) | 5500/7000 | Natural gas tax rate $T_{gas}$ (%) | 9 |
| Thermal coal tax rate $T_{coal}$ (%) | 13 | Natural gas transportation cost $TR_{gas}$ (Yuan/ton) | 300 |
| Coal transportation cost $TR_{coal}$ (Yuan/ton) | 40 | Natural gas transportation tax rate $T_{trgas}$ (%) | 9 |
| Coal transportation tax rate $T_{trcoal}$ (%) | 9 | The power consumption rate of gas-fired power plants $\sigma_{gas}$ (m$^3$/kWh) | 0.185 |
| Comprehensive coal consumption $\sigma_{coal}$ (g/kWh) | 320 | The electricity consumption rate of gas-fired power plants $\eta_{gafacelec}$ (%) | 2.35 |
| Water production cost $C_{wat}$ (fen/kWh) | 0.2 | Gas-fired power generation tax rate $T_{gaselec}$ (%) | 9 |
| Environmental cost $C_{env}$ (fen/kWh) | 0.2 | Fixed cost of gas-fired power generation $C_{gas}^{Const}$ (fen/kWh) | 7 |
| Coal-fired power generation tax rate $T_{coalelec}$ (%) | 13 | \ | \ |
| Fixed cost of coal-fired power generation $C_{coal}^{Const}$ (fen/kWh) | 10 | \ | \ |
| The power consumption rate of coal-fired power plants $\eta_{cofacelec}$ (%) | 5.5 | \ | \ |

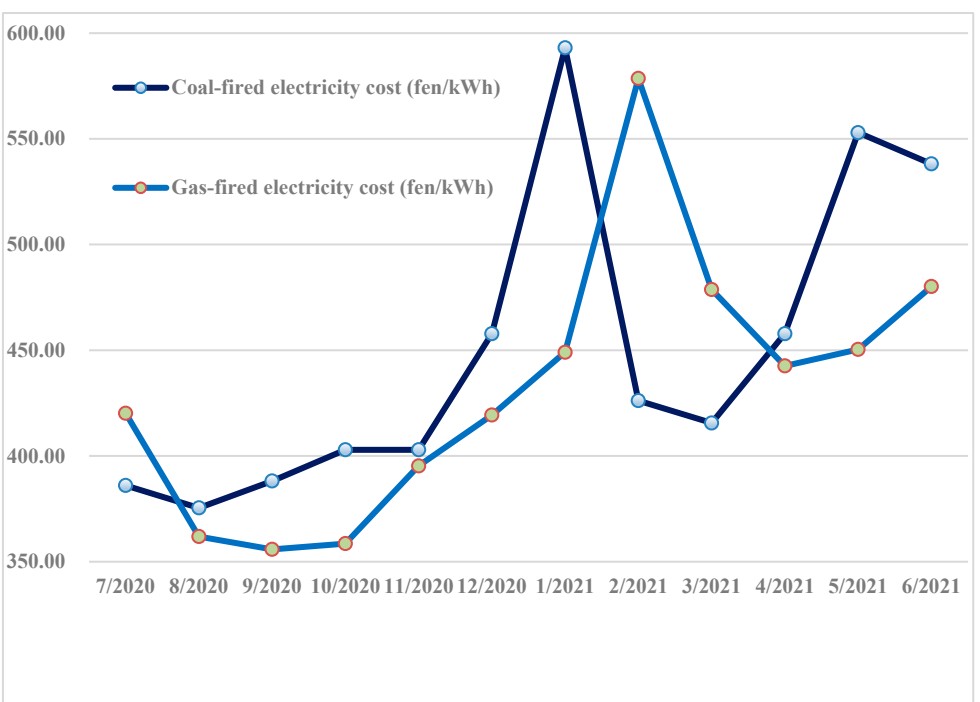

**Figure 5.** July 2020–June 2021 Trends in power generation costs of coal-fired and gas-fired power generation groups.

*5.2. Simulation Analysis of Power Generation Group Bidding Game Considering Primary Energy Price Fluctuation*

This section mainly takes the change in the marginal cost of power generation caused by the fluctuation of the primary energy price in the last section as the background and analyzes the evolutionary game equilibrium under the influence of multiple factors through several calculation examples. Parameters that remain the same for all of the following situations are the total demand for the electricity market $Q_{DM}$, the cost bidding coefficient of coal-fired power generation groups $\mu_{coal}$, the strategic bidding coefficient of coal-fired power generation groups $\nu_{coal}$, the cost bidding coefficient of gas-fired power generation groups $\mu_{gas}$, the strategic bidding coefficient of gas-fired power generation groups $\nu_{gas}$. These parameters are set as follows: $Q_{DM} = 10000$ MW, $\mu_{coal} = \mu_{gas} = 1.1$, $\nu_{coal} = \nu_{gas} = 1.3$.

Taking the trend of coal-fired and gas-fired power generation costs from July 2020 to June 2021, as described in Figure 5 as the background, first of all, in scenario 1, that is, the power generation of coal-fired and gas-fired power generators participating in the market is each smaller than the market demand, and the total power generation of coal-fired and gas-fired power generators participating in the market is greater than the market demand. The parameters are set to: $Q_{coal,1} \sim Q_{coal,5} = 1000$ MW; $Q_{coal,6} \sim Q_{coal,8} = 600$ MW; $Q_{coal,9} \sim Q_{coal,12} = 300$ MW; $Q_{gas,1} \sim Q_{gas,10} = 300$ MW; $Q_{gas,11} \sim Q_{gas,20} = 200$ MW.

In July 2020 and March 2021, the cost of gas-fired power generation is slightly higher than the cost of coal-fired power generation, which belongs to the scenario of phase 2 in Figure 2a. At this time, the bidding power of coal-fired and gas-fired power generation groups satisfies the first condition in Scenario 1.1-b, which is as stated in inequality (28):

$$\sum_{i=1}^{n} Q_{gas,i} < \frac{\nu_{gas} - 1}{\mu_{gas} - 1}(Q_{DM} - \sum_{i=1}^{m} Q_{coal,i}) \tag{28}$$

The evolution simulation results are shown in Figure 6a, the supply is relatively tight, and the coal-fired and gas-fired power generation groups all choose strategic bidding, which leads to the high electricity price. The coal-fired electricity price is $\nu_{coal}$ times the

cost of its power generation, the gas-fired electricity price is $\nu_{gas}$ times the cost of its power generation, and the market has reached an equilibrium situation. In order to alleviate the pressure of high electricity prices, the bidding volume of gas-fired power generation groups can be increased to satisfy the second condition in Scenario 1.1-b, which is as stated in inequality (29):

$$\frac{\nu_{gas}-1}{\mu_{gas}-1}\left(Q_{DM}-\sum_{i=1}^{m}Q_{coal,i}\right) < \sum_{i=1}^{n}Q_{gas,i} < Q_{DM}-\sum_{i=1}^{m}Q_{coal,i}\frac{\mu_{coal}-1}{\nu_{coal}-1} \qquad (29)$$

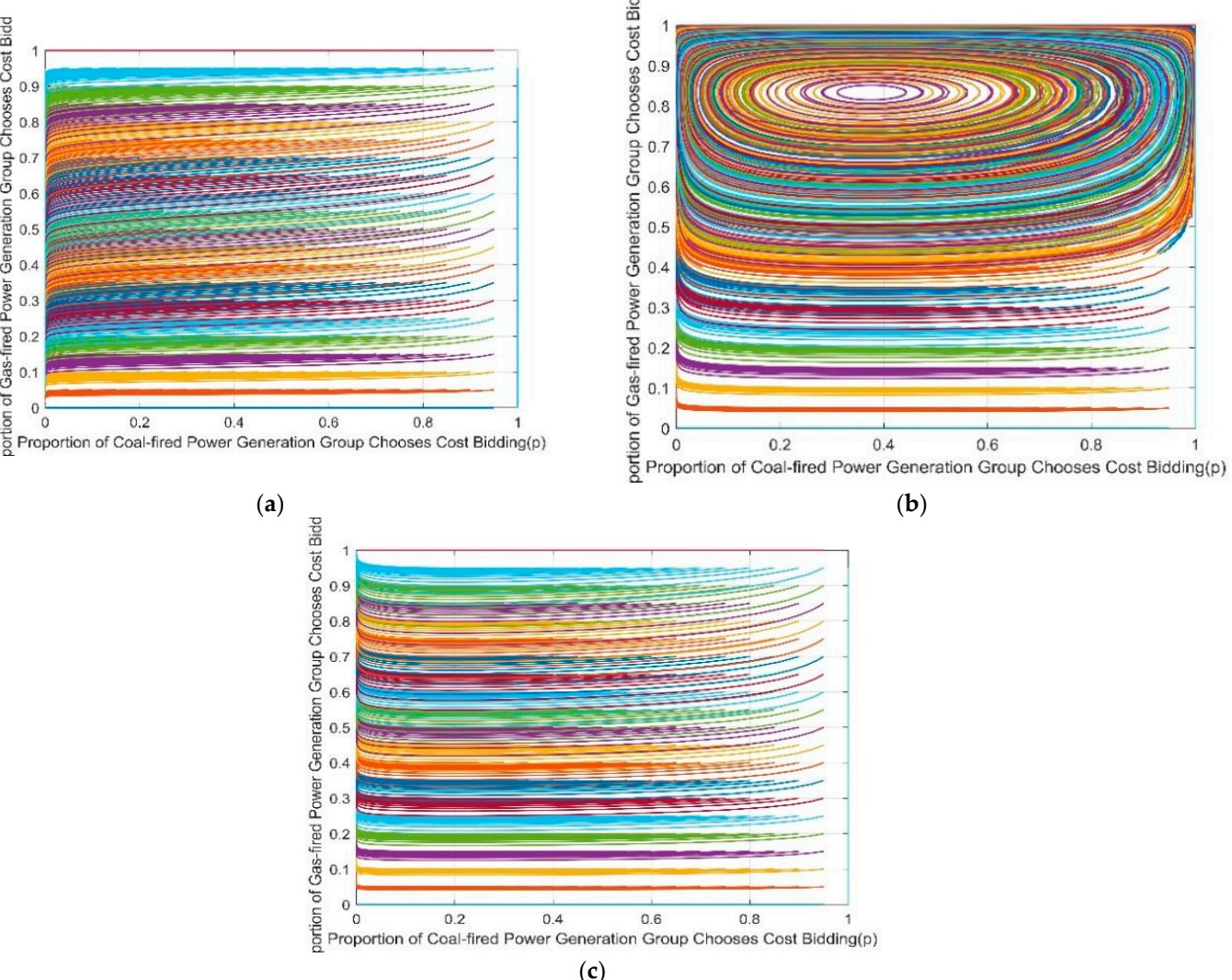

**Figure 6.** Evolution trend of bidding entities in phase 2 under scenario 1. (**a**) Simulation results of the evolution of coal-fired and gas-fired feed-in tariffs under "Condition 1" in Scenario 1.1-b, (**b**) Evolutionary simulation results under "Condition 2" in Scenario 1.1-b, (**c**) Simulation results for the evolution of "condition three" in scenario 1.1-b.

Let $Q_{gas,1} \sim Q_{gas,10} = 300$ MW; $Q_{gas,11} \sim Q_{gas,30} = 200$ MW. The evolution simulation results are shown in Figure 6b. The competition on the supply side has intensified. Gas-fired power generation groups that adopt strategic bidding can obtain higher unit income than gas-fired power generation groups that adopt cost bidding, but the total income decreases. After a long-term evolution, the relatively large amount of gas bidding power on the generation side cannot change the bidding behavior of the coal-fired power

generation groups, but it can change the bidding behavior of the gas-fired power generation groups. Therefore, the coal-fired power generation groups still adopt strategic bidding, and the gas-fired power generation groups adopt cost bidding, and the electricity price has dropped. Coal-fired electricity prices are still $v_{coal}$ times the cost of its power generation. However, the gas-fired electricity price has dropped by $\mu_{gas}$ times the cost of its power generation. The market can still reach equilibrium. Further, increase the bidding volume of gas-fired power generation groups to meet the third condition in Scenario 1.1-b, which is as stated in inequality (30):

$$\sum_{i=1}^{n} Q_{gas,i} > \max\left\{Q_{DM} - \frac{\mu_{coal}-1}{v_{coal}-1}\sum_{i=1}^{m} Q_{coal,i}, \frac{v_{gas}-1}{\mu_{gas}-1}\left(Q_{DM} - \sum_{i=1}^{m} Q_{coal,i}\right)\right\} \tag{30}$$

Let $Q_{gas,1} \sim Q_{gas,15} = 300$ MW; $Q_{gas,16} \sim Q_{gas,30} = 200$ MW. The evolution simulation results are shown in Figure 6c. The fierce competition on the power generation side has prompted the coal-fired and gas-fired power generation groups to make strategic bidding no longer the optimal behavior after long-term evolution. The market cannot reach an equilibrium state, two kinds of bidding behaviors coexist, and the electricity prices fluctuate. The coal-fired electricity price is $\mu_{coal} \sim v_{coal}$ times the cost of its power generation, and the gas-fired electricity price is $\mu_{gas} \sim v_{gas}$ times the cost of its power generation.

Next, the remaining phases are simulated under the supply relationship $Q_{coal,1} \sim Q_{coal,3} = 1000$ MW; $Q_{coal,4} \sim Q_{coal,7} = 600$ MW; $Q_{coal,8} \sim Q_{coal,13} = 300$ MW; $Q_{gas,1} \sim Q_{gas,5} = 300$ MW; $Q_{gas,6} \sim Q_{gas,20} = 200$ MW.

From August to December 2020 and April and June 2021, the cost of coal-fired power generation is slightly higher than the cost of gas-fired power generation, which belongs to the situation of phase 3 in Figure 2a. At this time, the coal-fired and gas-fired bidding volume satisfies the first condition in Scenario 1.2-a, which is as stated in inequality (31):

$$\sum_{i=1}^{m} Q_{coal,i} < \frac{v_{coal}-1}{\mu_{coal}-1}\left(Q_{DM} - \sum_{i=1}^{n} Q_{gas,i}\right) \tag{31}$$

The evolution simulation results are shown in Figure 7a. Increase the bidding volume of gas-fired power generation groups to meet the second condition in Scenario 1.2-a, which is as stated in inequality (32):

$$\frac{v_{coal}-1}{\mu_{coal}-1}\left(Q_{DM} - \sum_{i=1}^{n} Q_{gas,i}\right) < \sum_{i=1}^{m} Q_{coal,i} < Q_{DM} - \sum_{i=1}^{n} Q_{gas,i}\frac{\mu_{gas}-1}{v_{gas}-1} \tag{32}$$

Let $Q_{gas,1} \sim Q_{gas,20} = 300$ MW; $Q_{gas,21} \sim Q_{gas,31} = 200$ MW, the evolution simulation results are shown in Figure 7b. Increase the bidding volume of gas-fired power generation groups again to meet the third condition in Scenario 1.2-a, which is as stated in inequality (33):

$$\sum_{i=1}^{m} Q_{coal,i} > \max\left\{Q_{DM} - \frac{\mu_{gas}-1}{v_{gas}-1}\sum_{i=1}^{n} Q_{gas,i}, \frac{v_{coal}-1}{\mu_{coal}-1}\left(Q_{DM} - \sum_{i=1}^{n} Q_{gas,i}\right)\right\} \tag{33}$$

The conditional relations in scenario 1.2-b are shown in inequalities (34) and (35):

$$(\mu_{gas} - v_{gas})C_{gas} < \frac{Q_{DM} - \sum\limits_{i=1}^{n} Q_{gas,i}}{\sum\limits_{i=1}^{m} Q_{coal,i}}(\mu_{coal} - v_{coal})C_{coal} \tag{34}$$

$$(\mu_{gas} - \nu_{gas})C_{gas} > \frac{Q_{DM} - \sum\limits_{i=1}^{n} Q_{gas,i}}{\sum\limits_{i=1}^{m} Q_{coal,i}} (\mu_{coal} - \nu_{coal})C_{coal} \tag{35}$$

The evolution simulation results are shown in Figure 7c. The simulation process of phase 3 is similar to that of phase 2, and its analysis will not be repeated here.

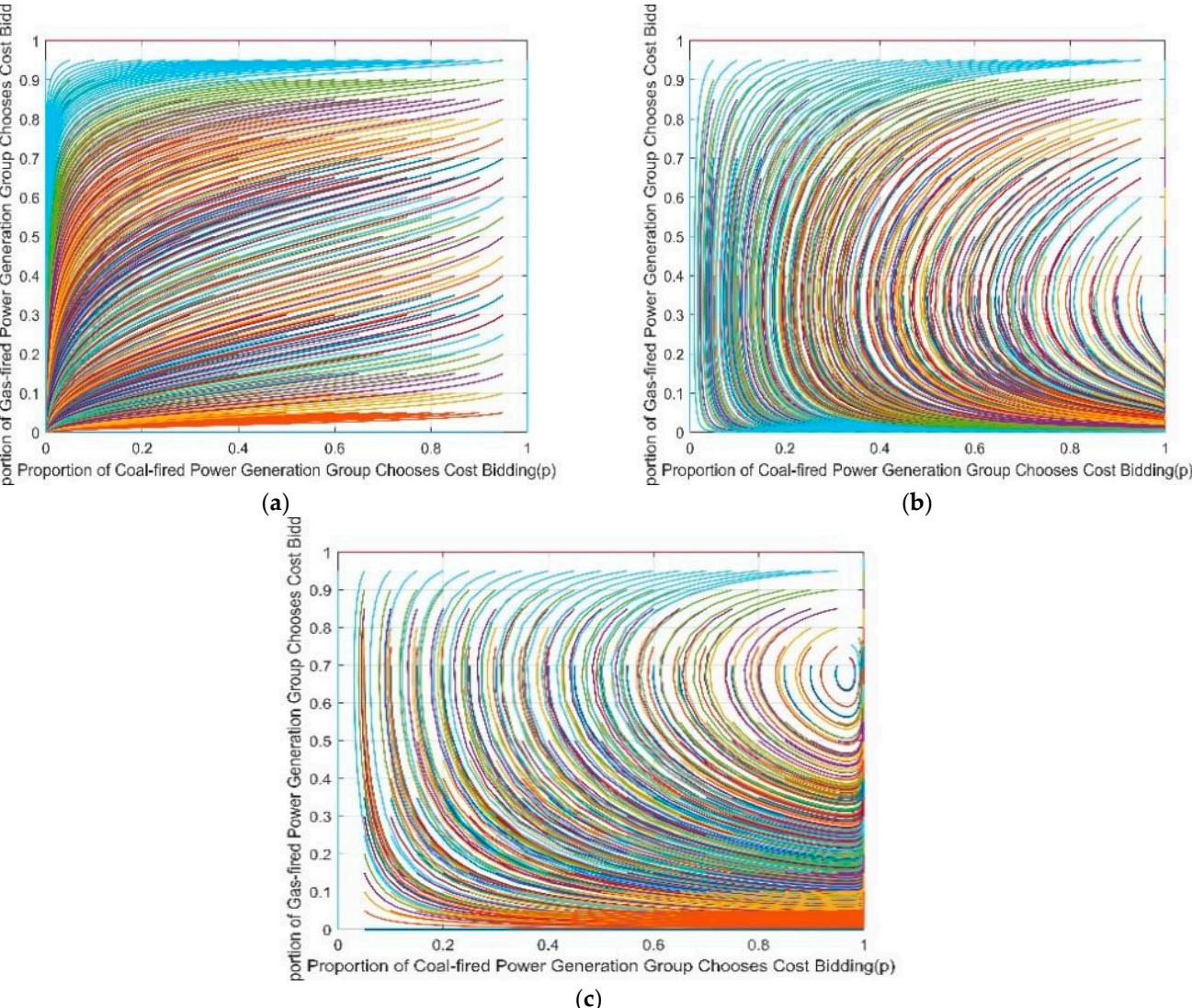

(a)

(b)

(c)

**Figure 7.** Evolution trend of bidding entities in phase 3 under scenario 1. (**a**) Simulation results of the evolution of coal-fired and gas-fired feed-in tariffs under "Condition 1" in Scenario 1.2-a, (**b**) Simulation results for increasing the feed-in tariff for gas-fired generators to satisfy "Condition 2" in Scenario 1.2-a, (**c**) Simulation results of increasing the feed-in tariffs of gas-fired generators again to satisfy "condition three" in scenario 1.2-a.

In February 2021, the cost of coal-fired power generation was much lower than the cost of gas-fired power generation, which belongs to the situation of phase 1 in Figure 2a, and the evolution simulation results are shown in Figure 8a. In January and May 2021, the cost of coal-fired power generation is much higher than that of gas-fired power generation, which belongs to the situation of phase 4 in Figure 2a, and the evolution simulation results are shown in Figure 8b. The large gap in energy costs leads to the that the party with higher power generation costs will not affect the bidding behavior of the party with lower power

generation costs. Both coal-fired and gas-fired power generation groups choose strategic bidding. The market has reached equilibrium, but the electricity price is too high, the coal-fired electricity price is $v_{coal}$ times the cost of its power generation, and the gas-fired electricity price is $v_{gas}$ times the cost of its power generation.

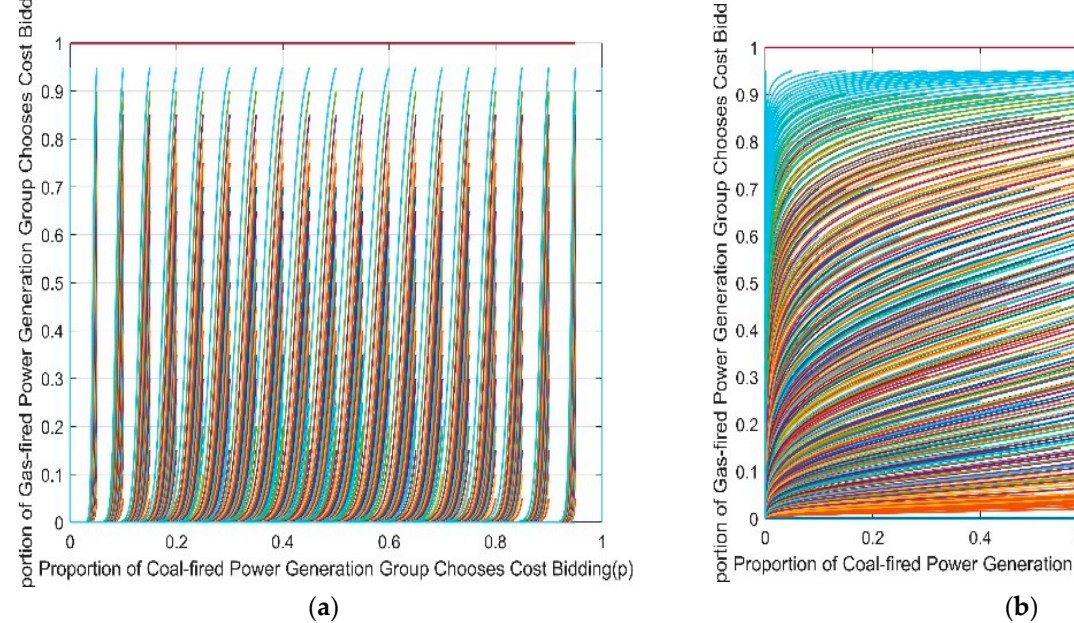

(a)                                    (b)

**Figure 8.** Evolution trend of bidding entities in phases 1 and 4 under scenario 1. (**a**) Evolutionary simulation results for February 2021 when coal-fired unit cost of electricity is much lower than gas-fired unit cost of electricity, (**b**) Evolutionary simulation results for January and May 2021 when the cost of coal-fired electricity is much higher than the cost of gas-fired electricity.

Secondly, the simulation verification is made under the supply-demand relationship in scenario 2. That is, the power generation of coal-fired power generators participating in the market is greater than the market demand, and the power generation of gas-fired power generators participating in the market is less than the market demand. The parameters are set to: $Q_{coal,1} \sim Q_{coal,6} = 1000$ MW; $Q_{coal,7} \sim Q_{coal,11} = 600$ MW; $Q_{coal,12} \sim Q_{coal,21} = 300$ MW; $Q_{gas,1} \sim Q_{gas,10} = 300$ MW; $Q_{gas,11} \sim Q_{gas,15} = 200$ MW.

In July 2020 and March 2021, the cost of gas-fired power generation was slightly higher than the cost of coal-fired power generation, which belongs to the situation of phase 2 in Figure 2b. At this time, the bidding power of coal-fired and gas-fired power generation groups satisfies the first condition in Scenario 2.1-b, which is as stated in inequality (36):

$$\sum_{i=1}^{n} Q_{gas,i} < Q_{DM} \frac{v_{coal} - \mu_{coal}}{v_{coal} - 1} \tag{36}$$

The evolution simulation results are shown in Figure 9a. The coal-fired power generation group chooses strategic bidding, and the gas-fired power generation group adopts cost bidding, which leads to the coal-fired electricity price being $v_{coal}$ times the cost of its power generation, the gas-fired electricity price is $\mu_{gas}$ times the cost of its power generation, the market has reached equilibrium situation. In order to reduce the electricity price, the bidding volume of gas-fired power generation groups can be increased to satisfy the second condition in Scenario 2.1-b: $\sum_{i=1}^{n} Q_{gas,i} < Q_{DM} \frac{v_{coal} - \mu_{coal}}{v_{coal} - 1}$, let $Q_{gas,1} \sim Q_{gas,20} = 300$ MW; $Q_{gas,21} \sim Q_{gas,30} = 200$ MW. The evolution simulation results are shown in Figure 9b. More gas-fired bidding volume forces the coal-fired power generation group to change their bidding behavior, that is, to adopt cost bidding. At this time, the large market power of the

coal-fired power generation group plus the low power generation cost makes the coal-fired power generation group occupy all of the power markets after long-term evolution. The gas-fired power generation group has no optimal bidding strategy, and the two bidding behaviors coexist. The electricity price is relatively low, which is $\mu_{coal}$ times the cost of coal-fired power generation.

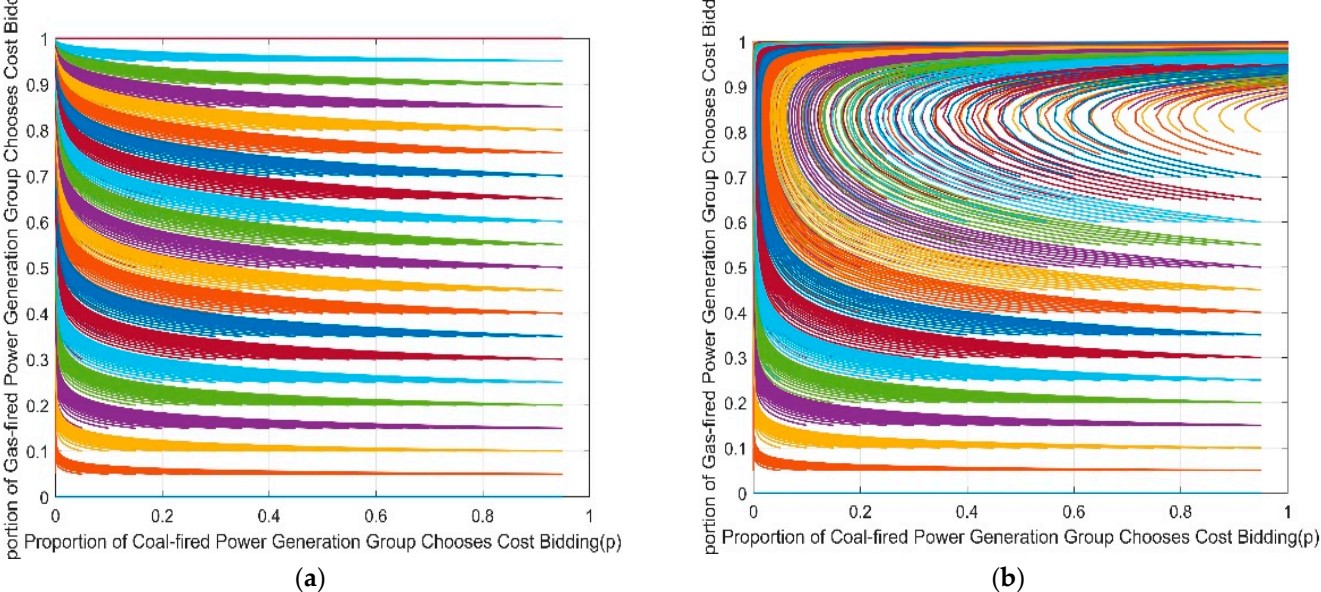

**Figure 9.** Evolution trend of bidding entities in phase 2 under scenario 2. (**a**) Simulation results of the evolution of gas-fired electricity cost in July 2020 and March 2021 when it is slightly higher than coal-fired electricity cost, (**b**) To reduce the price of electricity can increase the amount of gas-fired generators feed-in bids, the evolutionary simulation results at this point.

From August to December 2020 and April and June 2021, the cost of coal-fired power generation is slightly higher than the cost of gas-fired power generation, which belongs to the situation of phase 3 in Figure 2b. At this time, coal-fired and gas-fired on-grid bidding electricity satisfies the first condition in Scenario 2.2-a: $\sum_{i=1}^{n} Q_{gas,i} < Q_{DM}\frac{v_{coal}-\mu_{coal}}{v_{coal}-1}$, and the evolution simulation results are shown in Figure 10a. The coal-fired and gas-fired power generation groups all choose strategic bidding, which leads to high electricity prices. The coal-fired electricity price is $v_{coal}$ times the cost of its power generation, the gas-fired electricity price is $v_{gas}$ times the cost of its power generation, and the market has reached an equilibrium situation. In order to reduce the electricity price, the bidding volume of the gas-fired power generation group can be increased to satisfy the second condition in Scenario 2.2-a, which is as stated in inequality (37):

$$\sum_{i=1}^{n} Q_{gas,i} > Q_{DM}\frac{v_{coal} - \mu_{coal}}{v_{coal} - 1} \tag{37}$$

Let $Q_{gas,1} \sim Q_{gas,20} = 300$ MW; $Q_{gas,21} \sim Q_{gas,30} = 200$ MW. The evolution simulation results are shown in Figure 10b. The fierce competition in the power generation side market has prompted the coal-fired and gas-fired power generation groups to make strategic bidding no longer the optimal behavior after long-term evolution, and the electricity market cannot reach an equilibrium state. The coexistence of two bidding behaviors leads to the fluctuation of electricity prices, the coal-fired electricity price is $\mu_{coal} \sim v_{coal}$ times the cost of its power generation, and the gas-fired electricity price is $\mu_{gas} \sim v_{gas}$ times the cost of its power generation.

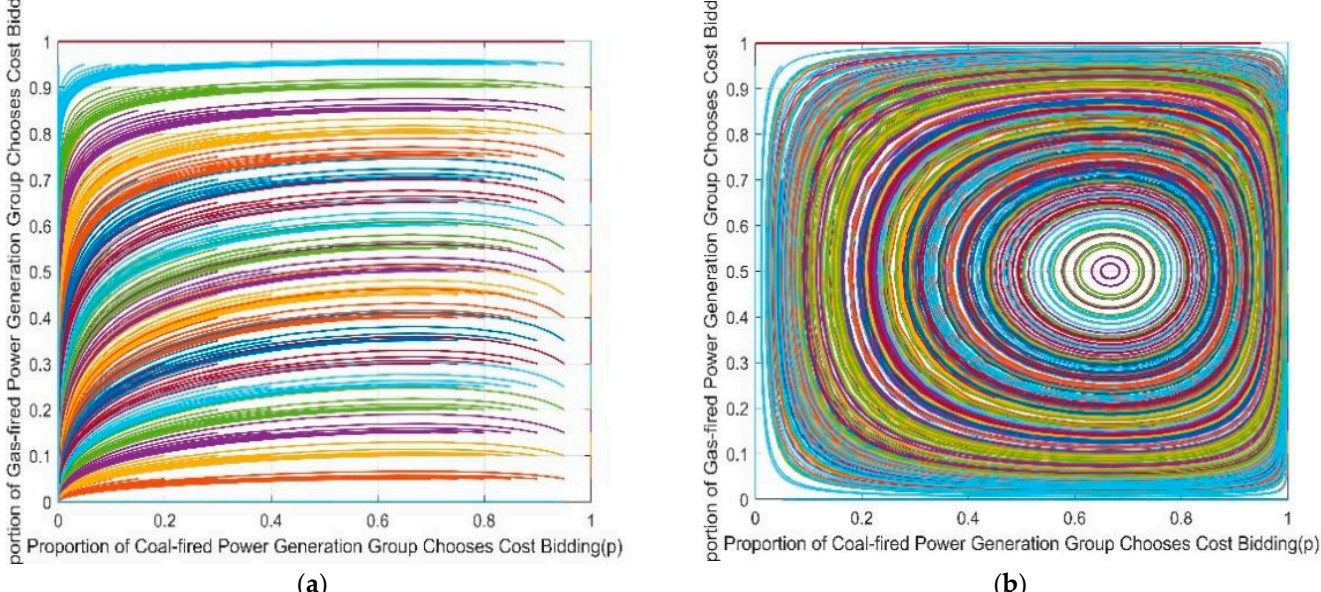

**Figure 10.** Evolution trend of bidding entities in phase 3 under scenario 2. (**a**) Simulation results of the evolution of the cost of coal-fired electricity from August to December 2020 and April and June 2021 with a slightly higher cost of electricity than gas-fired electricity, (**b**) Simulation results for the evolution of a gas-fired power producer with a higher feed-in tariff for lower electricity prices.

The conditional relations in scenario 2.2-b are shown in inequalities (38) and (39):

$$\frac{Q_{DM} - \sum\limits_{i=1}^{n} Q_{gas,i}}{\sum\limits_{i=1}^{m} Q_{coal,i}} C_{coal} > C_{gas} \tag{38}$$

$$\frac{Q_{DM} - \sum\limits_{i=1}^{n} Q_{gas,i}}{\sum\limits_{i=1}^{m} Q_{coal,i}} C_{coal} < C_{gas} \tag{39}$$

In February 2021, the cost of coal-fired power generation was much lower than the cost of gas-fired power generation, which belongs to the situation of phase 1 in Figure 2b, and the evolution simulation results are shown in Figure 11a. In this case, there is no stable equilibrium point, and the coal-fired power generation group chooses the strategic bidding, the gas-fired power generation group has a huge disadvantage in power generation cost, and the power generation side market has been completely occupied by the coal-fired power generation group, so the revenue is 0. The bidding strategy of the game subject cannot reach a stable equilibrium state, but the electricity price can reach a stable state, which keeps $v_{coal}$ times the cost of coal-fired power generation. In January and May 2021, the cost of coal-fired power generation is much higher than that of gas-fired power generation, which belongs to the situation of phase 4 in Figure 2b, and the evolution simulation results are shown in Figure 11b. Coal-fired and gas-fired power generators all choose strategic bidding. The market has reached equilibrium, but the electricity price is high. The coal-fired electricity price is $v_{coal}$ times the cost of its power generation, and the gas-fired electricity price is $v_{gas}$ times the cost of its power generation.

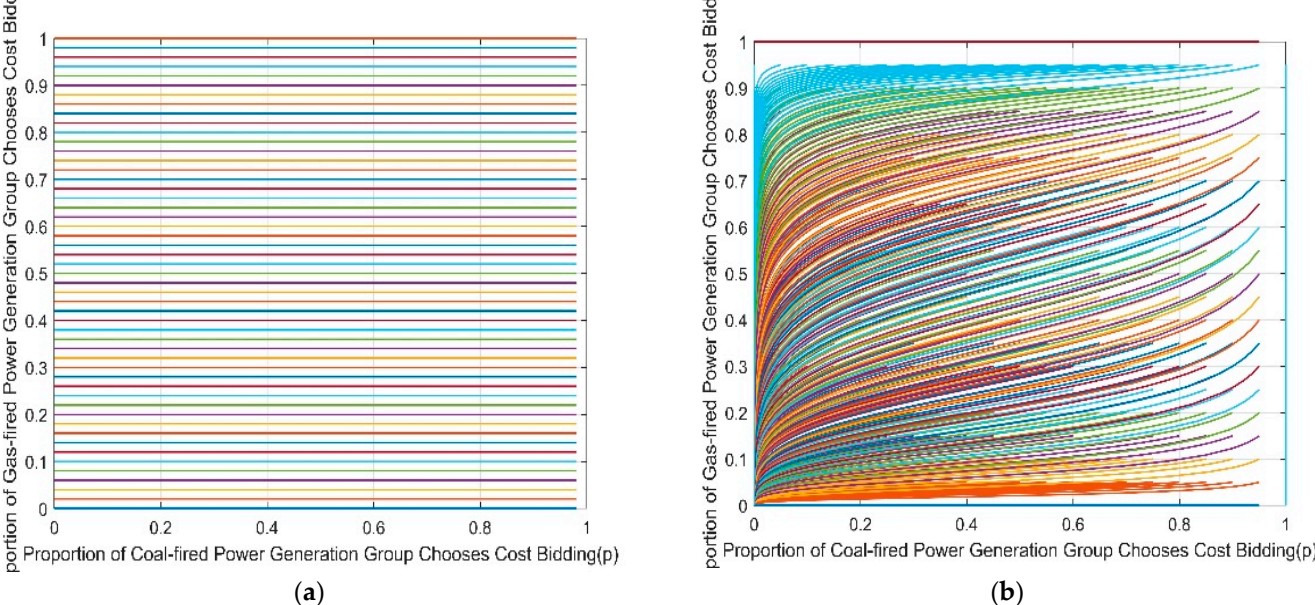

**Figure 11.** Evolution trend of bidding entities in phases 1 and 4 under scenario 2. (**a**) Simulation results of the evolution of the cost of coal-fired electricity at a much lower cost than gas-fired electricity in February 2021, (**b**) Evolutionary simulation results when the cost of coal-fired electricity is much higher than the cost of gas-fired electricity in January and May 2021.

In addition, according to the discussion in Section III and the parameter settings in Section V, the cost relationship between primary and secondary energy can be quantified as follows. First, the relationship between the cost of coal-fired electricity and the price of thermal coal can be expressed as Equation (40):

$$C_{coal} = 4.22 \times 10^{-4} \lambda_{coal} + 0.124 \tag{40}$$

Formula (26), $\lambda_{coal}$ is the price of thermal coal.

Secondly, the relationship between the cost of gas-fired electricity and the price of natural gas can be expressed as Equation (41):

$$C_{gas} = 1.35 \times 10^{-4} \lambda_{gas} + 0.11 \tag{41}$$

Formula (27), $\lambda_{gas}$ is the price of natural gas.

When a certain supply-demand relationship is met, same-stage bidding will not increase the overall electricity price. There is little difference in electricity cost. That is, the price based on cost quotation chosen by high-cost generators is lower than the price quoted by low-cost generators using strategy. Specifically, two types of generators can be represented in Equation (42):

$$\mu_{gas}/v_{coal} < C_{coal}/C_{gas} < v_{gas}/\mu_{coal} \tag{42}$$

When $\mu_{coal} = \mu_{gas} = 1.1$, $v_{coal} = v_{gas} = 1.3$, the cost ratio of coal and gas power generation can be obtained between 0.846 and 1.182. In the context of using primary energy prices, it can be quantified as Equation (43):

$$0.27\lambda_{gas} - 73.28 < \lambda_{coal} < 0.38\lambda_{gas} + 14.22 \tag{43}$$

If let $\lambda_{coal} = 600$, then when $1542 < \lambda_{gas} < 2492$, that is, when the cost ratio of coal and gas power generation is between 2.57 and 4.15. The bidding of gas and coal-fired power generation groups on the same stage will not lead to an increase in electricity prices, and the market can maintain a balanced state.

Summarizing the above calculation results, when the coal-fired power generation in the market cannot meet the load demand after the expiry, coal-fired units are gradually withdrawn from operation, that is, when the ratio of coal-fired to gas-fired power generation costs is between 0.846 and 1.182, if gas-fired power generators bidding with coal-fired power generators on the same stage, the electricity price in the entire market will be stable. Taking the thermal coal price of 600 yuan/ton as an example, according to Formulas (26)–(29), when the natural gas price is 1542~2492 yuan/ton, the gas and coal-fired power generation groups bidding on the same stage will not lead to electricity price rises. However, if the price of natural gas is outside this range, the market price of electricity will rise, and the market may deviate from the equilibrium state. At this time, the market management department should set up a market structure for gas-fired power generation and coal-fired power generation to bid separately.

## 6. Discussion

When the coal-fired power generation capacity can meet the load demand and the price of natural gas is higher than that of coal, gas-fired power generation should not enter the market to compete with coal-fired power generation at the same stage. Making it act as a peak-shaking unit or participating in an auxiliary service market is a scheme to keep the power market system running steadily and is conducive to market equilibrium. With the prohibition of new construction of coal-fired power generation and the gradual withdrawal of more and more expired coal-fired units, the capacity of coal-fired power generation will not fully support the power load at that time, and the gas-fired units will enter the market rationally to undertake the task of power supply. This paper deduces that when coal-fired power generation cannot meet the load demand in the market, under what conditions can the primary energy price ratio of natural gas and thermal coal meet, the market structure of coal-fired and gas-fired power generation groups bidding on the same platform can be set, which can meet the market equilibrium and will not push up the market price. Otherwise, coal-fired and gas-fired power generation groups can be set to bid separately, which can also achieve market equilibrium and maintain the stability of electricity prices.

The research will consider the situation after the new energy power generation participates in the market bidding and further consider the impact and evolution path of the volatility and uncertainty of new energy power generation on the future low-carbon power system.

## 7. Conclusions

Gas power generation is an inevitable trend in the development of future power generation models, and it is also an important way to meet the needs of environmental protection and economic development. How to design a reasonable market mechanism which can not only encourage gas-fired power generation to participate in the electricity market but also use it as a means of controllable adjustment under the condition of primary energy fluctuations to stabilize the market supply and demand balance, is a double-edged sword proposition. The core of the market mechanism is to study the opportunity and role of gas-fired power generation units entering the market for bidding.

Based on the evolutionary game theory applied to the analysis of group behavior, this chapter analyzes the changes in group power generation costs and bidding behavior characteristics of power producers with the fluctuation of primary energy prices to study the timing and role of gas-fired power generation entering the market. Based on the gas-fired and coal-fired power generation group evolution game model under the influence of multiple factors, and taking into account the power cost, bidding mechanism, the number of power generation groups, power generation capacity, supply-demand relationship and other factors, the equilibrium state of the power generation group game under various circumstances and the conditions that the equilibrium state needs to meet are analyzed, and the method for judging the stable equilibrium point of the market evolution game is proposed. By taking the actual price fluctuation of coal and natural gas as input data, the

example verifies the rationality of the model and the stable equilibrium discrimination method in this chapter.

**Author Contributions:** Y.J. contributed to the determination and verification of game simulation conditions. C.Q. contributed to the evolutionary game model design and simulation. J.Y. contributed to the design of the simulation scene. L.Z. and Q.C. contributed to the English editing and checking. Z.W. contributed to the collation of simulation data. Y.W. and X.M. contributed to the verification of the simulation scene and the acquisition of input data. All authors have read and agreed to the published version of the manuscript.

**Funding:** This research was funded by the State Grid Corporation Science and Technology Project "Analysis and key technology research on situation equilibrium and market bidding risk trend of provincial power market" grant number (No. J2021080).

**Data Availability Statement:** The data presented in this study are available in the article.

**Acknowledgments:** The authors thank the two reviewers whose comments improved the quality of this paper.

**Conflicts of Interest:** The authors declare no conflict of interest.

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
