# Peer review of "Evolutionary Game Analysis of Power Generation Groups Considering Energy Price Fluctuation"

_algorithms, doi:10.3390/a15120456_

Round 1

Reviewer 1 Report

A few specific comments (and suggestions):

1. The paragraph on page 3 starting in line number 117: "Based on ..." Second sentence "According to the quotation tendency.." is very convoluted and hard to understand, if not incorrect. Please rewrite. Especially "the management department of the power market adjusts the online bidding power of the two.." does not make any sense, and is not a description of any energy market I am aware of.

2. The above specific comment relates to a more general comment that the institutional set up with energy/electricity markets is not very well described and outlined. Thus it is hard to see how well this evolutionary game fits with actual markets. A better description of the  link between actual markets and the evolutionary game analyzed here would improve the paper.

3. The exact nature of the "strategic bidding" strategy is not explained, and the reader is left wondering if this is just a label attached to a (rather arbitrary) strategy distinct from the cost-based bidding strategy. It would be helpful to have a clear explanation how this strategy has been derived, or how it is evolving over time.

4. The paper uses the term "market supply exceeds demand". In an electricity market it is a design requirement that supply exactly equals demand in any outcomes. Please explain and rewrite.

5. The conclusion relates the evolutionary game model to the design of a market mechanism. The underlying model is not sufficiently anchored in a market design to allow for a discussion of mechanism design. As stated above, the institutional details are not there.

Author Response

Point 1: The paragraph on page 3 starting in line number 117: "Based on ..." Second sentence "According to the quotation tendency.." is very convoluted and hard to understand, if not incorrect. Please rewrite. Especially "the management department of the power market adjusts the online bidding power of the two.." does not make any sense, and is not a description of any energy market I am aware of. 

Response 1: Relevant parts in the paper have been modified accordingly.

Point 2: The above specific comment relates to a more general comment that the institutional set up with energy/electricity markets is not very well described and outlined. Thus it is hard to see how well this evolutionary game fits with actual markets. A better description of the  link between actual markets and the evolutionary game analyzed here would improve the paper.

Response 2: In the revised version, the statement on the institutional setting of energy market and electricity market has been added.

Point 3: The exact nature of the "strategic bidding" strategy is not explained, and the reader is left wondering if this is just a label attached to a (rather arbitrary) strategy distinct from the cost-based bidding strategy. It would be helpful to have a clear explanation how this strategy has been derived, or how it is evolving over time.

Response 3: The authors add the definition and connotation of "strategic bidding" strategy..

Point 4: The paper uses the term "market supply exceeds demand". In an electricity market it is a design requirement that supply exactly equals demand in any outcomes. Please explain and rewrite.

Response 4: Relevant parts in the paper have been modified accordingly.

Point 5: The conclusion relates the evolutionary game model to the design of a market mechanism. The underlying model is not sufficiently anchored in a market design to allow for a discussion of mechanism design. As stated above, the institutional details are not there.

Response 5: In the revised version, the author added a supplement to the market mechanism. The results of the evolutionary game process are the reference for the design of the market mechanism, and verify whether the design of the market mechanism is reasonable.

Reviewer 2 Report

This is an interesting paper which employs evolutionary game theory and studies the bidding behavior on the power generation domain.

Considering multiple features such the power cost, bidding mechanism, the number of power generation groups, power generation capacity, supply-demand relationship etc. to analyze the equilibrium state of the power generation seems interesting. It would be an interesting addition to see the weightage each feature brings in.

Author Response

Point 1: Considering multiple features such the power cost, bidding mechanism, the number of power generation groups, power generation capacity, supply-demand relationship etc. to analyze the equilibrium state of the power generation seems interesting. It would be an interesting addition to see the weightage each feature brings in. 

Response 1: There are relatively few evolutionary game studies considering the multiple characteristics of power cost, bidding mechanism, number of generation groups, generation capacity, supply and demand relationship, etc. The comments of reviewers will be the focus of the author's future research, but this article does not focus on this aspect.

Reviewer 3 Report

The article deals with the possible application of evolutionary games in the problem of analysis and trading on energy markets. The paper studies a specific selected case and performs an extensive data analysis. However, despite the considerable efforts of the authors and access to valuable data, which must be positively evaluated, in my opinion, the paper is not ready for acceptance and publication. The most significant shortcomings are as follows:

- Lack of a clear definition of originality. What is the contribution of knowledge in the field of algorithms, or at least trading and analysis of time series and volatile markets?

- There is a lack of comparison with any other "bidding" technique to use as a baseline and to define the benefits based on this comparison clearly.

- The article is extremely difficult to read, and the presentation of the data is often vaguely explained/discussed.

- The information presented in the graphs is not explained. For example, what do the different colors in Figures 6-11 or the arrows in Figure 1 mean?

- Presentation of data in tables is very unclear - tables contain words with capital letters UNKNOWN across several lines, navigating the table is difficult even with accompanying text. The equations in Fig. 2 are redrawn with a different graphic (why are they in the figure at all).

- Equation 7 lacks symbols.

- Formal comments - the text is difficult to read, the equations are disorganized, and many of them are not referenced in the text itself. State of the art contains repetitive sentences "Reference XX..." The abstract contains sentences that lack verbs. Sometimes the conjunction something-based occurs, and then again, the word based

- Lack of analysis of differences, and improvements over the article in reference 11.

- I generally miss the definition of "algorithm."

Author Response

Point 1: Lack of a clear definition of originality. What is the contribution of knowledge in the field of algorithms, or at least trading and analysis of time series and volatile markets? 

Response 1: The scope of Algorithms includes “Algorithmic game theory and mechanism design”, etc. The core of this paper is the application of evolutionary game model in generation side bidding of electricity market. The evolutionary path and game equilibrium of evolutionary game under multi - scenario and multi - initial conditions are discussed.

Point 2: There is a lack of comparison with any other "bidding" technique to use as a baseline and to define the benefits based on this comparison clearly.

Response 2: This paper does not propose a new bidding strategy, but simulates the game equilibrium state of the market as a whole when the market subject adopts different bidding strategies, so as to judge whether the declared electricity on the generation side and the declared electricity on the consumer side of the market are balanced.

Point 3: The article is extremely difficult to read, and the presentation of the data is often vaguely explained/discussed.

Response 3: Some intermediate data in this paper, such as the simulation results of evolutionary game path, are not the core of this paper, so there is no detailed discussion.

Point 4: The information presented in the graphs is not explained. For example, what do the different colors in Figures 6-11 or the arrows in Figure 1 mean?

Response 4: The author has added explanations of relevant contents in the revised version. The arrows in Figure 1 represent the evolution path. The different colors in Figures 6-11represent the evolution path under different initial conditions.

Point 5: Presentation of data in tables is very unclear - tables contain words with capital letters UNKNOWN across several lines, navigating the table is difficult even with accompanying text. The equations in Fig. 2 are redrawn with a different graphic (why are they in the figure at all).

Response 5: The “UNKNOWN” of the chart presentation has been optimized to some extent.Figure 2 shows the distribution of all scenarios in the example analysis, including information such as the value range of variables. It can be considered as a multi scenario regional distribution map divided by the value range of variables.

Point 6: Equation 7 lacks symbols.

Response 6: According to the author's verification, equation 7 seems to have no problem.

Point 7: The text is difficult to read, the equations are disorganized, and many of them are not referenced in the text itself. State of the art contains repetitive sentences "Reference XX..." The abstract contains sentences that lack verbs. Sometimes the conjunction something-based occurs, and then again, the word based

Response 7: Relevant parts in the paper have been modified accordingly.The author adds quotations and explanations to the equations. In the literature review part, the description of references is modified.

Point 8: Lack of analysis of differences, and improvements over the article in reference 11.

Response 8: The classic evolutionary game model is described in Reference 11. The contribution of this paper is to model the bidding problem of different types of generator units in the power market, and then use the evolutionary model theory to conduct group simulation, which expands the application of evolutionary game model in complex environments, and analyzes the game path and game equilibrium in multiple scenarios.

Point 9: I generally miss the definition of "algorithm."

Response 9: This paper designs and verifies the electricity market bidding mechanism according to the evolutionary game path and the evolutionary game results. This solution itself is also an algorithm.

Round 2

Reviewer 3 Report

The authors have made minor changes to the article, but it is still not acceptable. Here are the main comments:

- the level of English is extremely poor. A simple check of the Abstract in the Grammarly application (there is also a free basic version, but checked in the profi EDU version) found 11 critical errors in the Abstract and 13 errors in the first paragraph of the introduction. The first sentence of the Abstract does not make sense. It does not have the correct grammar and is basically missing a verb. The third sentence begins: "Based on the evolutionary game..." again has no verb. The results of the check are in the attached pdf to this review. Without rigorous proofreading, the article cannot be easily understood.

- The authors have explained in their responses to comments what the basic idea, the possible originality, is in this article...but why is it not identifiable in the article itself? If they spent some time writing a response, why didn't they put it in the paper? There needs to be a subsection of the "Introduction" section titled, for example, "Originality and motivation," where the originality, contribution, and explained differentiation to similar work (like ref 11 already discussed) is clearly explained.

- The referencing of equations in the text is still done vaguely. The reader is completely lost in the text. All numbered equations must be referenced in the text, but not in the way the authors did in the corrected version (by adding the phrase, "according to the following equation"). Every sentence that mentions and refers to a relationship, condition, or equation must clearly reference which one. Example: line 175. For a population dynamic described by a system of differential equations, the... Which system? Add ref... and there are dozens of similar issues.

- I do not understand the reply, that some results are not the core of the paper. Why are they there in the first place? But of course, it can attract other researchers (more complex studies). Maybe add a better explanation in the introduction section on how the paper is organized, a common standard in scientific papers.

- Completely redraw/rebuild the figure 2. It is highly unprofessional and unacceptable to insert an image into a scientific journal where the equation is unreadable and redrawn with a different graphic. Why, for example, for the top left corner: scenario 1 phase 1, is a simple reference to Condition 1 and a reference to equation (XX) not inserted in the graphics box? Why are the equations from the text repeated again in this graphic and often unreadable?

- Formal footnote: I would like to politely remind the Authors that as a reviewer, I have access to all versions of the paper, and from them, I can see that there were indeed symbols missing in equation 7, thus I do not understand the response to the comment that there is no problem with the equation (see attached pdf).

Author Response

Response to Reviewer 3 Comments

Firstly, the authors would like to express our sincere apologies to the reviewer for his/her generosity in pointing out problems in the submitted academic paper. Due to compatibility issues, there may have been some misunderstandings. The authors have made changes in response to the reviewer's comments and apologize again.

Point 1: The level of English is extremely poor. A simple check of the Abstract in the Grammarly application (there is also a free basic version, but checked in the profi EDU version) found 11 critical errors in the Abstract and 13 errors in the first paragraph of the introduction. The first sentence of the Abstract does not make sense. It does not have the correct grammar and is basically missing a verb. The third sentence begins: "Based on the evolutionary game..." again has no verb. The results of the check are in the attached pdf to this review. Without rigorous proofreading, the article cannot be easily understood. 

Response 1: The authors are grateful to the reviewers for raising issues of English grammar and writing conventions in the paper. The authors have double-checked the writing issues throughout the revised version.

Point 2: The authors have explained in their responses to comments what the basic idea, the possible originality, is in this article...but why is it not identifiable in the article itself? If they spent some time writing a response, why didn't they put it in the paper? There needs to be a subsection of the "Introduction" section titled, for example, "Originality and motivation," where the originality, contribution, and explained differentiation to similar work (like ref 11 already discussed) is clearly explained.

Response 2: The authors are very grateful to the reviewers for their suggestions and have added explanatory notes on the originality, contributions, and discrepancies of the paper in the revised version.

Point 3: The referencing of equations in the text is still done vaguely. The reader is completely lost in the text. All numbered equations must be referenced in the text, but not in the way the authors did in the corrected version (by adding the phrase, "according to the following equation"). Every sentence that mentions and refers to a relationship, condition, or equation must clearly reference which one. Example: line 175. For a population dynamic described by a system of differential equations, the... Which system? Add ref... and there are dozens of similar issues.

Response 3: The author has added the necessary explanatory notes to the quotations in important parts of this paper.

Point 4: I do not understand the reply, that some results are not the core of the paper. Why are they there in the first place? But of course, it can attract other researchers (more complex studies). Maybe add a better explanation in the introduction section on how the paper is organized, a common standard in scientific papers.

Response 4: The authors would like to thank the reviewer for the suggested changes to the paper, and we have added notes on the structure and Language organization of the paper in the revised version.

Point 5: Completely redraw/rebuild the figure 2. It is highly unprofessional and unacceptable to insert an image into a scientific journal where the equation is unreadable and redrawn with a different graphic. Why, for example, for the top left corner: scenario 1 phase 1, is a simple reference to Condition 1 and a reference to equation (XX) not inserted in the graphics box? Why are the equations from the text repeated again in this graphic and often unreadable?

Response 5: The authors are very grateful to the reviewers for the suggested changes, and in the revised version the authors have redrawn Figure 2. The unreadability of the equations is also most likely related to the version and compatibility of Mathtype.

Point 6: Formal footnote: I would like to politely remind the Authors that as a reviewer, I have access to all versions of the paper, and from them, I can see that there were indeed symbols missing in equation 7, thus I do not understand the response to the comment that there is no problem with the equation (see attached pdf).

Response 6: The authors would like to begin by expressing their sincerest apologies to the reviewers. The formulae in this paper were edited using Mathtype software version 7.4.4.516, which may have caused some symbols to be displayed incompletely due to software compatibility issues, but they are displayed in full on our devices. 

Round 3

Reviewer 3 Report

I would definitely like to thank the Authors for their exemplary handling of the comments and their honest responses. As it stands, all my comments have been incorporated and I see no obstacles to accepting this article. Once again, before the actual production process, it would be a good idea to check grammar and typos.